# A Comparative Analysis of the Polyphenolic Content and Identification of New Compounds from *Oenothera biennis* L. Species from the Wild Flora

**DOI:** 10.3390/molecules30204059

**Published:** 2025-10-11

**Authors:** Viviane Beatrice Bota, Neli-Kinga Oláh, Elisabeta Chișe, Ramona-Flavia Burtescu, Flavia-Roxana Pripon Furtună, Lăcrămioara-Carmen Ivănescu, Maria-Magdalena Zamfirache, Endre Máthé, Violeta Turcuș

**Affiliations:** 1Doctoral School of Biology, Faculty of Biology, “Alexandru Ioan Cuza” University of Iași, 700505 Iasi, Romania; ivanescu@uaic.ro (L.-C.I.); magda@uaic.ro (M.-M.Z.); 2Faculty of Medicine, “Vasile Goldiș” Western University of Arad, 310414 Arad, Romania; endre.mathe@agr.unideb.hu; 3National Institute for Economic Research “Costin C. Kirițescu”, Romanian Academy, Centre for Mountain Economy (CE-MONT), 725700 Suceava, Romania; 4SC PlantExtrakt SRL, Rădaia, 407059 Cluj, Romania; neliolah@yahoo.com (N.-K.O.); ramona.burtescu@plantextrakt.ro (R.-F.B.); flavia.pripon@plantextrakt.ro (F.-R.P.F.); 5Faculty of Pharmacy, “Vasile Goldiș” Western University of Arad, 310414 Arad, Romania; chise.elisabeta@uvvg.ro; 6Institute of Nutrition, Faculty of Agricultural and Food Sciences and Environmental Management, University of Debrecen, H-4032 Debrecen, Hungary

**Keywords:** *Oenothera biennis* L., evening primrose, phytochemistry, polyphenols, flavonoid, phenolic acid, Romania

## Abstract

*Oenothera biennis* L. is a biennial species native to North America and introduced to Europe in the 17th–18th centuries, used in traditional medicine or as a dietary supplement in various products, as well as in cosmetics and the pharmaceutical industry. In recent decades, oil extracted from seeds has been increasingly used for the treatment of various conditions. In this article, we highlight the polyphenolic content of 2 types of extracts from *O. biennis* species, collected from the wild flora of Romania, from two regions with different altitudes and pedoclimatic conditions (plains and mountains), namely: OHM—hydroalcoholic extract, Macea sample; OHVD—hydroalcoholic extract, Vatra Dornei; OAM—aqueous extract, Macea sample; OAVD—aqueous extract, Vatra Dornei sample. The LC/MS analysis of the whole plant extracts revealed 5 to 14 polyphenols, depending on the sample location and type of extract, out of which 7 flavonoids are newly reported for this species. Climatic parameters were mapped using QGIS. Higher qualitative and quantitative values of polyphenols were observed in the hydroalcoholic extract obtained from individuals collected from the plain area.

## 1. Introduction

*Oenothera biennis* L. is a herbaceous species belonging to the Onagraceae Juss family [1]. The common name “evening primrose” refers to the flowers that open at sunset [2], while “fever plant” and “king’s cure-all” recall the period when the plant was used to treat fever and considered a universal remedy [3].

The species is endemic to North America, with a range that extends from Mexico to Florida, north to central Ontario and Quebec (Canada), west to North Dakota and Oklahoma, and along the Pacific coast [2]. It was subsequently introduced to all continents except Antarctica [2], being the most common and widespread species in the *Oenothera* subsection [4]. Its spread in Europe originates from two distinct groups of *Oenothera* spp. Native to America [4].

In Romania, *O. biennis* is an alien, pioneer species that grows in sandy areas or on skeletal soils, on the edges of roads and railways, on riverbanks, gravel beds, near walls, in ruderal areas, and rarely in cultivated areas, in the *Dauco-Sisymbrion*, *Melilotion*, and *Onopordetalia* plant associations [5,6].

The earliest uses of the species are recorded in the medicinal and dietary practices of Native Americans. Currently, the most widely used part of the plant is the seeds, from which the oil rich in essential fatty acids, especially gamma-linolenic acid (GLA), is extracted and marketed as a dietary supplement or ingredient in various products (cosmetics, pharmaceuticals, etc.) [7,8,9].

A comprehensive review of the extant literature was conducted, encompassing 50 studies published prior to October 2025, which focus on the phytochemical composition of *O. biennis* L. (Appendix A) [10,11,12,13,14,15,16,17,18,19,20,21,22,23,24,25,26,27,28,29,30,31,32,33,34,35,36,37,38,39,40,41,42,43,44,45,46,47,48,49,50,51,52,53,54,55,56,57,58,59,60]. Research on this species has mainly centered on the composition of seeds and products derived from them (meal, cakes), germinated seeds/sprouts, and seed oil/evening primrose oil (EPO), accounting for 29 studies. The interest in EPO is due to the presence in its composition of omega-6 essential fatty acids, linoleic acid, and gamma-linolenic acid, along with other bioactive compounds such as triterpenes, phenolic acids, tocopherols, and phytosterols [61]. A smaller number of studies (24) highlight the phytochemical composition of the leaves, roots, aerial parts, and the whole plant, while an initial mention of some glycosylated flavonoids does not mention the plant part utilized. The fractions of free and hydrolyzed phenolic acids (acidic and alkaline) in the herb and seeds, respectively, have been highlighted as containing salicylic acid, p-hydroxybenzoic acid, protocatechuic acid, vanillic acid, gentisic acid, p-coumaric acid, caffeic acid, ferulic acid, p-hydroxyphenylacetic acid, syringic acid, gallic acid, pyrocatechuic acid, 2-hydroxy-4-methoxybenzoic acid, and gamma-resorcinol [23,24]. Flavonoids have been confirmed in the leaves [15,29,62] and whole plant [34]. The hydroethanolic extract from sprouts contains miquelianin, luteolin-7-glucuronide, and quercetin-3-O-glucuronide [59,60]. Other flavonoids isolated from seeds were trihydroxyflavanone derivatives, quercetin-xylopyranoside, kaempferol-glucuronide, quercetin, trimethyleneglycol-digallate, monogalloylglucose, digalloylglucose, protocatechuic acid, procyanidin B, procyanidin trimer, (+)-catechin, methyl-gallate, procyanidin B gallate, catechin-gallate [52]. Tannins have been reported in seed meal [54] and leaves [15], and have recently been confirmed in various extracts from the whole plant [34]. Several authors have identified the presence of ellagitannins—oenothein A and B, oligomers of oenothein—in leaves [12,16,17,18,63], and in the stem and roots [64]. The presence of oenothein B in the aerial parts was highlighted in the aqueous extract [25], respectively, in the crude extract obtained from filtered and lyophilized methanolic extract [26]. Alkaloids have been reported in the leaves [15] and in aqueous, ethanolic, and ethyl acetate extracts of the whole plant [34]. The seeds and the oil extracted from them contain various sterols, 4-methylsterols [35], dimethylsterols [38], phytosterols [35,42,43], 4-desmethylsterol, erythrodiol, and uvaol [51]. Sterols were also found in leaves [14,15], and the roots present oenotheralanosterol A and B of particular medicinal value [31,32].

Other compounds reported in the composition of *O. biennis* are: cetyl alcohol, paraffin, resin, phlobaphenes, tannins, pentosans, mucilage in leaves [5], 2-methyl-7-oxo-tritetracont-1, 5-dien-2l-ol [29], dihydroxyprenylxanthone, cetoletyl diglucoside, oenotheaphytilactone, oenotheraphenoxylactone in the root [31,32], tetralin lactones, dodecenyl-benzene-triol, prenyl-anthracene-diol, acyl-diglucoside, and benzoic acid [33], saponins in aqueous, ethanolic, and ethyl acetate extracts from the whole plant [34]. The hydromethanolic extract from the leaves contains: methyl ester 10-octadecenoic acid, 4H-1-bezopyran-4-one-7-hydroxy-2-(4-hydroxyprenyl), 2,6-bis(1,1-dimethyl)-4-[(4-hydroxy-3,5-dimethylphenyl)methyl]-phenol, 4-methyl-1-(1-methylethyl)-3-cyclohexen-1-ol, 3-Buten-2-one, 4-(2,5,6,6-tetramethyl-2-cyclohexan-1-yl), 4,8,12,16-Tetramethylheptadecan-4-olide, and isopropyl stearate [21]. In addition to sugars and amino acids, Wang et al. (2021) [52] highlight the presence of minerals, oxalic acid, and citric acid in seeds.

The scientific literature highlights the biological action of EPO and GLA obtained from seeds in the treatment of diabetes and related conditions. EPO has shown positive effects on the fatty acid profile in patients with type 1 diabetes, influencing several parameters analyzed by Van Doormaal et al. (1988) [65]. The effect of EPO on the circulatory system is often presented in the context of diabetes mellitus pathology, using various in vivo experimental models [66,67,68,69,70,71,72,73,74,75,76,77]. Several studies show that EPO increases HDL cholesterol levels and reduces serum triglyceride levels [66,68,69,72,78,79,80,81]. Additionally, other parts of the plant have also demonstrated biological activity that is relevant in the treatment of conditions such as cancer, as evidenced by studies conducted over the last decade. A water-soluble polysaccharide with antitumor effects was isolated from the whole plant [82]. Oenotheralanosterol B, isolated from the root in combination with oenotheralanosterol A, has shown in vitro antiproliferative activity on liver, breast, prostate, leukemia, and mouse macrophage cell lines [32]. The extract from the aerial parts has antiproliferative and proapoptotic activity on the human melanoma cell line A357 [27]. In vitro, the stem extract stopped the proliferation and migration of vascular smooth muscle cells by regulating proteins involved in the cell cycle [83].

In this paper, we aim to perform a comparative analysis and biochemical characterization of the hydroalcoholic and aqueous extracts from whole *O. biennis* plants. Considering the increasing interest in how environmental factors influence the phytochemical composition of plants, we collected samples from populations growing in two regions with different ecological conditions (plains and mountains), from Romania’s wild flora, and analyzed the LC/MS results in relation to altitude and pedoclimatic factors. Based on GIS spatial analysis, the spatial distribution and variability of climatic parameters collected from both site regions over the two-year development period of the species were mapped.

## 2. Results

Geographical location and biotope conditions are reported to influence the biochemical composition of plants, resulting in different therapeutic properties [84,85]. Therefore, altitude is an important component that introduces variability and influences the main environmental factors [86,87,88].

For the present study, the *O. biennis* species was collected from two distinct sites in the wild flora of Romania, located in plain and mountain regions, respectively. The analysis of climatic factors from each site area was conducted over two years, considering the biennial nature of the plant.

The first biological samples were collected from Macea, in western Romania. The site is located at an altitude of 99 m, and belongs to the northern forest-steppe zone with *Quercus robur*. From a biogeographical perspective, it corresponds to the Pannonian region. From a climatic perspective, it corresponds to the temperate-continental zone with oceanic influences [89,90]. The other samples for our study were collected from Vatra Dornei in northeastern Romania at an altitude of 926 m. This site belongs to the nemoral zone, the subzone of beech forests (*Fagus sylvatica*) and mixed beech and coniferous forests (*Abies alba*). Biogeographically, the area corresponds to the Alpine region, and climatically, it falls within the cold continental zone with Scandinavian-Baltic influences [89,90].

In the Macea area, the average annual temperature was 11.7–12.2 °C in the harvest year (2021), and 12.2–12.7 °C in the first year of growth (2020), according to regional weather stations. Similarly, in the Vatra Dornei area, the average annual temperature was between 0.7 and 9.8 °C in 2021 and between 1.9 and 10.5 °C in 2020 (Figure 1).

In the harvest year of 2021, cumulative precipitation in Macea reached values between 395.3 and 578 mm, with higher values recorded by the stations closest to the locality throughout the year. In 2020, during the early stages of vegetation, cumulative precipitation ranged from 412.9 to 603.8 mm. Higher precipitation values than in the previous year were recorded at the station in Șiria. In the Vatra Dornei area, cumulative precipitation ranged from 757.4 to 1528.1 mm in 2021 and from 691.4 to 1419.5 mm in 2020 (Figure 2).

In terms of soil conditions in the harvesting areas, there are two distinct types of soil: one specific to lowland areas, mainly represented by typical chernozem/black soil with a basic pH, and one specific to mountainous areas, brown soil, which has a characteristic acidic pH. This may also influence the phytochemical profile of the species under study.

Chromatographic analysis of hydroalcoholic and aqueous extracts obtained from whole *O. biennis* plants collected from the wild flora of Romanian plains and mountains revealed the presence of between five and 14 polyphenols, depending on the site of origin and the type of extract. According to the scientific literature to date presented in Appendix A, seven flavonoids were identified for the first time in this species.

The LC/MS analysis of the hydroalcoholic extract allowed the identification and quantification of 14 polyphenols in the sample collected from Vatra Dornei (OHVD), and 13 polyphenols in the sample collected from Macea (OHM) (Table 1).

Quantitatively, the main compound in the hydroalcoholic extract was esculetin, in both sites, with the maximum concentration being in the mountain sample (OHVD)—1959 ± 46.3 μg/g dry plant weight. Additionally, OHVD recorded higher concentrations of chlorogenic acid, kaempferol, luteolin, naringenin, and rutoside compared to the plain sample (OHM) (Figure 3). In OHM, other quantitatively important compounds include hyperoside, present at a concentration approximately six times higher than OHVD, quercetin, present at a concentration almost three times higher, and salicylic acid (Figure 3). Apigenin was detected only in the OHVD extract, with a concentration of 10 ± 0.6 μg/g dry plant weight.

A total of 10 polyphenols were identified and quantified in the aqueous extract from the whole *O. biennis* plants in the Macea sample, and 5 in the Vatra Dornei sample. Overall, the quantities were lower than those found in the hydroalcoholic extract, as outlined in Table 2 and Figure 4 and Figure 5.

The presence of caffeic acid, esculetin, hesperetin, naringenin, quercetin, and rutoside was observed only in the aqueous extract from the plain sample (OAM), while chrysin was observed only in the aqueous extract from the mountain sample (OAVD).

From a quantitative perspective, the major compound in the OAM extract is hyperoside, with a concentration of 627 ± 15.6 μg/g dry plant weight, followed by esculetin, naringenin, salicylic acid, and rutoside. In the OAVD extract, the main compound remains hyperoside, but in a quantity approximately 4 times lower than in the OAM extract, followed by chlorogenic acid (137 ± 3.2 μg/g dry plant weight), salicylic acid (109 ± 2.7 μg/g dry plant weight) and chrysin (102 ± 2.6 μg/g dry plant weight) (Figure 4). However, hyperoside concentration is higher in OAVD than in the OHVD extract (Figure 5).

## 3. Discussion

Our findings are consistent with previous reports highlighting the influence of environmental and geographical factors on secondary metabolite accumulation in medicinal plants, where altitude, soil composition, and climatic variability have been shown to modulate both the qualitative and quantitative expression of bioactive compounds [91,92,93].

Regarding climatic factors, temperatures remained relatively constant during the two years of development of the collected plants (2020–2021), with a slight decrease in 2021, in both sites. Concerning precipitation, there was an increase in the mountainous area and a decrease in the plains area in the second year. Therefore, the conditions at the plains collection site were closer to the species’ ecological preferences (thermophilic, meso-xerophilic) [90,94].

The LC/MS analysis has revealed up to 14 phenolic acids and flavonoids in the hydroalcoholic extract from *O. biennis* plants collected from the mountain area (OHVD). Only the presence of 2 phenolic acids and 3 flavonoids was retained in the aqueous extract of the same collection site. In regard to the samples from the plain area, 13 of the same polyphenols were confirmed in the hydroalcoholic extract (OHM) and 10 in the aqueous extract (OAM). Seven of the flavonoids identified in this study have not been mentioned in previous publications regarding the phytochemical composition of *O. biennis* species: apigenin, chrysin, esculetin, hesperetin, hyperoside, luteolin, and naringenin. Our present report is primarily aligned with the scientific studies outlined in Appendix A [10,11,12,13,14,15,16,17,18,19,20,21,22,23,24,25,26,27,28,29,30,31,32,33,34,35,36,37,38,39,40,41,42,43,44,45,46,47,48,49,50,51,52,53,54,55,56,57,58,59,60].

It is important to note that the polyphenols discussed in this paper were identified in extracts obtained from multiple whole plants of the species sampled from two different populations in distinct sites, and that distribution and concentration may also vary with regard to the plant organ. In alignment with previous publications, caffeic acid has been identified in leaves [11,27], seeds [47] and seed meal [55]; chlorogenic acid in seeds [95], neochlorogenic acid in leaves [11]; salicylic acid in herba [23,24] and seeds [24,39]; gallic acid in leaves [11], herba [23,24,27], seeds and seed derivatives [24,39,40,52,54,55,57,60], and sprouts [47]; kaempferol and quercetin in leaves [11,13], herba [26], and seeds [52]; rutoside in herba [27]. Previous reports of compounds identified in this study are also summarized in Appendix A.

In addition, there are differences between the hydroalcoholic extract obtained by us from the whole plant in Romania and the sonicated hydroalcoholic extract from the herb obtained by Fecker et al. (2020) [27] from Tunisia. The latter contained higher quantities of caffeic acid, gallic acid, and rutoside, which supports the idea that geographic and pedoclimatic conditions influence this aspect.

While luteolin-7-glucuronide was previously mentioned in the sprout extract, it is a derivative of the luteolin that we are currently reporting in *O. biennis* [96].

The lower quantities of esculetin and flavonoid aglycones such as apigenin, luteolin, and quercetin found in the aqueous extract compared to the alcoholic extract can be explained by their lower solubility in water. In contrast, phenolic acids, chlorogenic acids, and gallic acids have similar solubility in water and ethyl alcohol.

Although the results of the hydroalcoholic extracts indicate a higher number of polyphenols in the mountain sample (OHVD), the overall quantities are lower than in the plain sample (OHM) (Figure 5). According to a 2018 study, the higher number of polyphenols can be explained by the more pronounced stress factors in cold climates [97], *Oenothera biennis* being a species usually associated with a warm climate, similar to that of Macea. However, the higher concentration of polyphenols in the plain sample is in contrast to the observations made by Anstett et al. (2018) [97].

Apigenin was detected in the lowest amount in the OHVD sample (10 µg/g dry plant), and in the OHM sample, it was below the quantification limit, suggesting a possible absence of the compound. This flavonoid is an antioxidant, antimicrobial, and antifungal, playing a role in increasing plant resistance to UV radiation and microbial/fungal infections. In other species, it is present in leaves, fruits, and flowers, and its concentration increases during leaf development [98,99].

The sample from Macea (OHM) contains a higher amount of quercetin (642 ± 16.0 µg/g dry plant) than OHVD (209 ± 5.2 µg/g dry plant), although this flavonoid provides better protection against UV radiation than apigenin [100]. It is possible that in the case of *O. biennis* plants from Vatra Dornei, apigenin production compensates for the lower level of quercetin, and that the change in metabolism and the promotion of the synthesis of other secondary metabolites may be a mechanism of adaptation to mountain stress factors and soil type, associated with the phenomenon of cross-tolerance [84].

Both samples showed a maximum concentration of esculetin, far exceeding that of the other compounds, with the highest amount found in the OHVD sample (1959 µg/g dry plant). This coumarin has numerous pharmacological activities, making it a therapeutic candidate for treating conditions such as diabetes and cancer [101]. According to Tattini et al. (2014) [102], esculetin accumulates in the vacuoles of palisade cells, preferentially in the adaxial portion, playing a role in photoprotection. The synthesis of esculetin coupled with increased peroxidase activity is considered a secondary antioxidant system in conditions of excessive light and depletion of the primary antioxidant system. In the OHM sample, in significant amounts, but less than half the amount of esculetin (1632 ± 38.5 µg/g dry plant), we detected hyperoside (732 ± 18.3 µg/g dry plant), quercetin (642 ± 16.0 µg/g dry plant), and salicylic acid (347 ± 8.7 µg/g dry plant). Hyperoside is a flavonoid-galactoside that plays a role in the plant’s response to UV radiation, drought, cold, and salinity stress, as well as reproduction [103]. The role of salicylic acid in plants has been well studied, as it is involved in defense against multiple stress factors and in the reproduction process [104]. In our study, the amount of these three compounds was considerably lower in the OHVD sample, where we note the presence of approximately 200 µg/g dry plant of quercetin, rutoside, and salicylic acid, followed by naringenin (154 µg/g dry plant) and hyperoside (116 µg/g dry plant).

Compared to the hydroalcoholic extract, we observed higher amounts of chlorogenic acid in the aqueous extract (137 ± 3.2—OAVD > 135 ± 3.3—OAM > 40 ± 1.0—OHVD > 14.6 ± 0.81—OHM). The amount of gallic acid was similar between the two types of extract (109 ± 2.7—OHM and 102 ± 2.6—OAM; 64 ± 1.7—OAVD and 61 ± 1.6—OHVD). Chrysin was present in a quantity of 102 µg/g dry plant in OAVD, and below the detection limit in the OAM sample, compared to 12 and 19 µg/g dry plant in the OHVD and OHM extracts, respectively. A similar situation was observed for hesperetin, naringenin, and rutoside, but this time the compounds were below the detection limit in the aqueous extract of the sample from Vatra Dornei (OAVD) and in higher amounts in the OAM sample. Naringenin is a flavonoid capable of inducing tolerance to abiotic stress factors and osmotic stress by regulating nitrogen metabolism and antioxidant effects [105].

As presented above, the newly identified compounds are well-known in the scientific literature for their medicinal properties and role in plant interactions with environmental factors. Therefore, their presence in *O. biennis* L. indicates the therapeutic potential of the species. Moreover, we observed differences in the polyphenolic content of hydroalcoholic and aqueous extracts, as well as between the two sites with characteristic plain and mountain pedoclimatic conditions. The quantitative and qualitative values were higher for hydroalcoholic extracts from *O. biennis* plants originating from low-altitude areas with a warm climate, high average annual temperatures, low cumulative precipitation, and basic soil, such as in the Macea area.

## 4. Materials and Methods

### 4.1. Collection of Vegetal Material

The vegetal material consists of whole *Oenothera biennis* L. plants collected from the wild flora of Romania, during their flowering period in August 2021 (in the afternoon), from the following locations:Vatra Dornei, Romania; altitude 926 m; GPS coordinates 47° 20′ 30.2136′′ N, 25° 19′ 55.0128′′ E; the samples were registered with voucher number 1256 in the Herbarium of the Plant Systematics discipline, part of the Department of Biology and Life Sciences at the “Vasile Goldiș” Western University of Arad (Romania) (Appendix A);Macea, Romania; altitude 99 m; GPS coordinates 46° 23′ 17.6676′′ N, 21° 18′ 7.7934′′ E; the samples were registered with voucher number 1257 in the Herbarium of the Plant Systematics discipline, part of the Department of Biology and Life Sciences at the “Vasile Goldiș” Western University of Arad (Romania) (Appendix A).

Several individuals belonging to the same population were collected from each location, with one specimen per population preserved as a voucher, and the rest used for the preparation of extracts. Vouchers correspond to the analyzed lots and were identified by Assoc. Prof. Dr. Turcuș Violeta (Appendix A). The number of individuals collected per location varied depending on availability and plant biomass. Collection was carried out in compliance with the legislation, research regulations in traditional medicine [106,107,108], and scientific literature recommendations regarding weather conditions at the time of sampling—warm weather, no precipitation, and ontogenetic stage—at maturity, respectively, during the flowering period [109].

### 4.2. Preparation of the Hydroalcoholic Extract

For the preparation of the hydroalcoholic extract, the whole plant was dried in the shade at room temperature for 7 days. The dried and crushed plant material was extracted in 70% alcohol at a ratio of 1:5 mass/volume [110,111]. The extract was obtained after 10 days of maceration with daily shaking for 20 min/day at 700–900 rpm, followed by vacuum filtration using W-type analytical filter paper for qualitative determinations, at room temperature, protected from light. The filtrate was brought to the initial volume for all samples. The ethanol used was 70% vol. concentration, obtained from 96% vol. ethanol (Coman Prod, Ilfov, Romania) and purified water (Millipore, Merck, Burlington, MA, USA). The final extraction ratio was 0.2 g dry plant/mL extract. The extraction yield was 20 mg/g dry plant. Concentrations counting the extraction ratio were back-calculated from the extract to the dry plant.

### 4.3. Preparation of the Aqueous Extract

For the preparation of the aqueous extract, the whole plant was dried for 7 days in the shade at room temperature. The extraction was performed by infusion, at a ratio of 1:10, of dried and crushed plant mixed with hot water (brought to a boil) [112]. The vessel was covered and the mixture was left to infuse for 30 min, then filtered using W-type analytical filter paper for qualitative determinations. The final extraction ratio was 0.1 g dried plant/mL extract. The extraction yield was 10 mg/g dry plant. Concentrations counting the extraction ratio were back-calculated from the extract to the dry plant.

### 4.4. Liquid Chromatography—Mass Spectrometry Analysis

The LC/MS analysis of plant extracts was performed on a Shimadzu Nexera I LC/MS-8045 UHPLC system (Kyoto, Japan), equipped with a quaternary pump and an autosampler, respectively, with an ESI source and a quadrupole mass spectrometer. Separation was performed on a Luna C18 reverse phase column (150 mm × 4.6 mm × 3 µm, 100 Å) from Phenomenex (Torrance, CA, USA). During analyses, the column was maintained at 40 °C. The mobile phase (Appendix A) consisted of a gradient of methanol (Merck, Darmstadt, Germany) and ultrapure water prepared using the Simplicity Ultra Pure Water Purification System (Merck Millipore, Billerica, MA, USA). Formic acid (Merck, Darmstadt, Germany) was used as an organic modifier. Methanol and formic acid were of LC/MS quality. The flow rate used was 0.5 mL/minute, with a total analysis time of 35 min. Detection was performed using a quadrupole mass spectrometer operated with electrospray ionization (ESI), both in negative and positive MRM (multiple reaction monitoring) ionization mode (Appendix A). The interface temperature was set at 300 °C. Nitrogen at 35 psi and 10 L/min was used for vaporization and as a carrier gas. The capillary potential was set at +3000 V. The standards are represented by the substances mentioned in Appendix A. From each standard, at each concentration, 1 µL was injected. Each sample/extract was diluted 1 mL to 10 mL with methanol, and aliquots of 10 μL were injected into LC/MS. The diluted samples were filtered through 0.45 μm PFTE membrane filters. Identification was performed by comparing the MS spectra and specific transitions between the separated compounds and standards. Identification and quantification were obtained based on the main transition in the MS spectra of the substances. The main transition was used for quantification, and both the main and secondary transitions were used for identification. For quantification purposes, calibration curves were also determined. Appendix A provides the calibration curve equations, their correlation coefficients, and the determined detection and quantification limits. Appendix A includes the retention times and specific transitions of the MS spectra specific to the standards used. Appendix A show retention times of the standards in parallel with those of the corresponding components separated from the studied extracts. Appendix A show the chromatograms, while Appendix A show the MS spectra.

To determine the matrix effect, standards were added to each extract so that the added concentration was twice that determined in the extract. LC/MS analyses were performed identically to the initial analyses. The same calibration curves were used for the calculations as for the initial determinations. Recovery was determined as the percentage of the ratio of the concentration determined in the sample with standard addition and the standard addition concentration. The recoveries obtained were between 148 and 151%, which corresponds to twice the concentration added to the samples. The matrix effect was determined as the percentage of the ratio between the concentration of the spiked sample and the sum of the concentrations of the unspiked sample and the concentration of the added standard. The percentages obtained range between 99.5 and 100.5%. Therefore, the matrix effect is very low and can be considered insignificant. The determinations for the matrix effect were performed in triplicate, and the recovery and matrix effect percentages mentioned are calculated as the average of the three determinations. The relative standard deviation of the three determinations for each sample and each separate and quantified component is below 10%.

“ND” is defined as “none detected” or “under detection limit”. “<QL” is defined as a component that can be identified (at a concentration greater than the detection limit), but which cannot be quantified within a reasonable error limit.

### 4.5. Spatial Modelling and Thematic Mapping of Climatic Parameters

In order to assess the climatic conditions in the sampling regions (plains and mountains), meteorological data were collected and analyzed from the Meteomanz database [113] from four weather stations close to the collection site for the years 2020–2021, thus encompassing the entire two-year development period of the plant. For the collection site in Macea, the data correspond to those issued by the weather stations in Chișineu-Criș (97 m—46° 31′ N/21° 32′E, Station code 15136); Arad (117 m, 46° 08′ N/21° 21′ E, Station code 15200), Șiria (474 m, 46° 15′ N/21° 39′ E, Station code 15179) and Sânicolaul-Mare (86 m, 46° 04′ N/20° 36′ E, Station code 15199). For collection site in Vatra Dornei, the data correspond to those issued by the weather stations in Poiana Stampei (924 m—47° 19′ N/25° 08′ E, Station code 15069), Călimani-Rețitiș (2021 m, 47° 05′ N/25° 14′ E, Station code 15088), Bistrița (367 m, 47° 08′ N/24° 30′ E, Station code 15085) and Iezer (1787 m, 47° 36′ N/24° 38′ E, Station code 15033).

The spatial distribution of climatic parameters, namely average annual temperature and cumulative precipitation, in the vicinity of the harvesting sites was modelled using QGIS 3.40 “Bratislava” [114]. For this purpose, the Inverse Distance Weighting (IDW) interpolation method [115,116] was applied to datasets obtained from the nearest meteorological stations, a technique that generates estimates for unsampled locations by weighting nearby values inversely with distance, thereby producing continuous raster surfaces that reflect spatial gradients in climate conditions [117,118].

The IDW method was selected as it provides a reliable representation of local climatic variability, particularly in heterogeneous environments such as mountain and lowland areas, where climatic contrasts significantly influence ecological processes. This approach has been shown to support robust environmental modelling and mapping, offering a balance between computational simplicity and interpretative accuracy in climate-ecology interactions [119,120].

The interpolated outputs were subsequently transformed into thematic maps, developed according to established cartographic principles, and integrated into GIS applications that combine environmental, territorial, and socio-cultural data [121,122,123,124]. This approach follows key developments in GIScience and applied studies that highlight the usefulness of spatial analysis for understanding landscape dynamics and sustainability issues [125,126,127,128].

### 4.6. Bibliographic Study

The literature review presented in Appendix A was conducted by consulting several types of resources, including original scientific articles, books, book chapters, abstracts, patents, and conference papers, available on major databases such as Web of Science, PubMed, ResearchGate, and Google Scholar. Various combinations of the following keywords were used: *Oenothera biennis*, phytochemistry, chemical composition, polyphenols, apigenin, chrysin, esculetin, hesperetin, hyperoside, luteolin, naringenin, caffeic acid, chlorogenic acid, salicylic acid, gallic acid, kaempferol, quercetin, and rutoside. The bibliographic study encompassed all database search results published before 6 October 2025, including non-English sources (such as Romanian, German, and Japanese with abstract translation). Although review articles were also consulted, only publications reporting original research results were included in Appendix A.

### 4.7. Statistical Analysis

The quantitative values of the compounds identified in the hydroalcoholic and aqueous extracts represent the mean ±standard deviations of three independent measurements of replicate injections of the same extract. Statistical analysis was performed using Excel software from the Microsoft Office package.

## 5. Conclusions

The analysis of the polyphenolic content of *O. biennis* species collected from two different regions of the wild flora of Romania has revealed quantitative and qualitative differences between the two sites characterized by distinct pedoclimatic factors (plain and mountain), and between the two types of extract (hydroalcoholic and aqueous). We report for the first time in *O. biennis* the presence of 7 flavonoids: apigenin, chrysin, esculetin, hesperetin, hyperoside, luteolin, and naringenin. In general, the polyphenol content of the hydroalcoholic extract is superior to that of the aqueous extract in terms of both quantity and quality. The polyphenol content in the extract obtained from plain area samples (Macea, Romania) is higher compared to that obtained from mountain area samples (Vatra Dornei, Romania). Notable differences were recorded in the content of chlorogenic acid, esculetin, kaempferol, luteolin, and naringenin in the hydroalcoholic extract of *O. biennis* collected from the mountain areas. The amount of gallic acid was similar in both types of extract. Future studies involving multiple sites, larger samples over a longer period, and soil analysis will be necessary to better understand the correlations between ecological factors and polyphenol variability in *O. biennis* plants. Furthermore, a comparative analysis of the polyphenolic content across different plant organs would provide insights into their distribution and primary accumulation sites within this species.

## Figures and Tables

**Figure 1 molecules-30-04059-f001:**
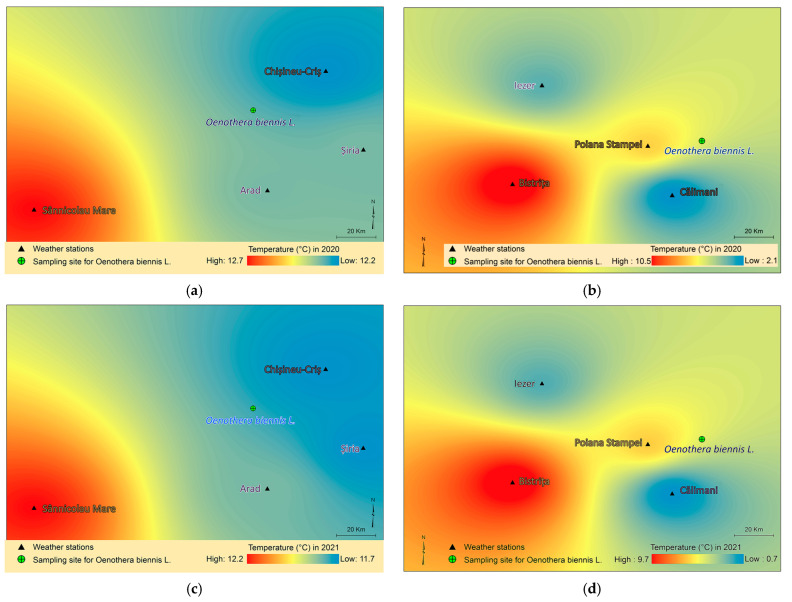
Maps of average annual temperatures based on data from four weather stations near the harvesting locations for the years 2020–2021. (**a**)—Macea 2020, (**b**)—Vatra Dornei 2020, (**c**)—Macea 2021; (**d**)—Vatra Dornei 2021.

**Figure 2 molecules-30-04059-f002:**
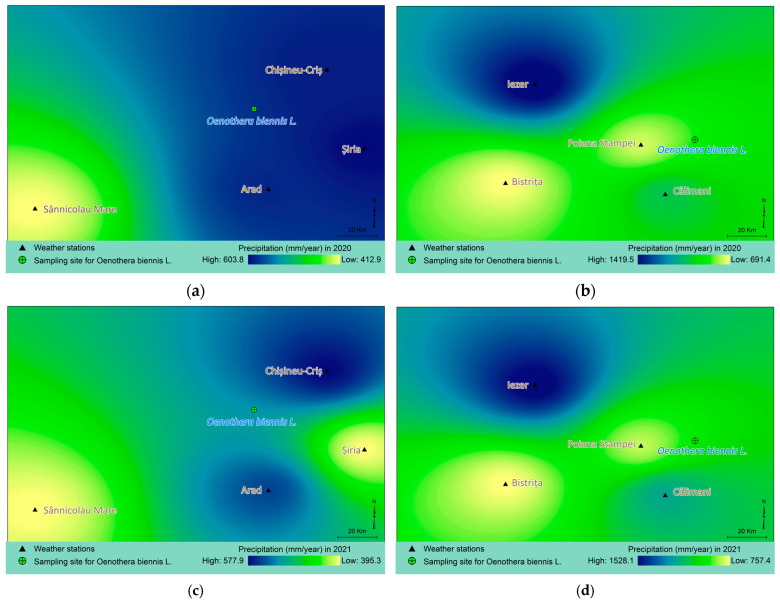
Maps of cumulative annual precipitation based on data from four weather stations near the harvesting locations for the years 2020–2021. (**a**)—Macea 2020, (**b**)—Vatra Dornei 2020, (**c**)—Macea 2021, (**d**)—Vatra Dornei 2021.

**Figure 3 molecules-30-04059-f003:**
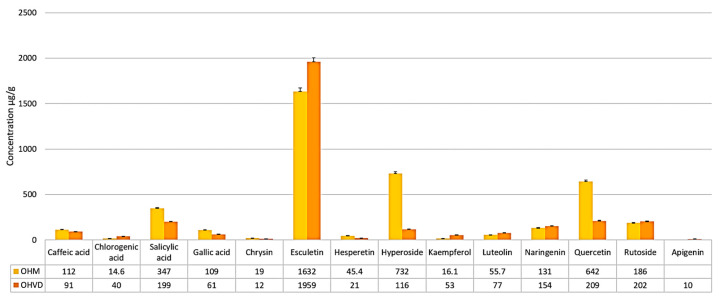
Graphical representation of phenolic acids and flavonoids identified in the hydroalcoholic extract of *O. biennis* collected from Macea (OHM) and Vatra Dornei (OHVD).

**Figure 4 molecules-30-04059-f004:**
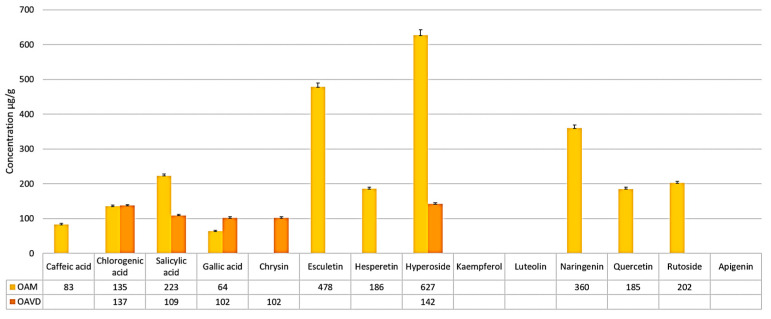
Graphical representation of phenolic acids and flavonoids identified in the aqueous extract of *O. biennis* collected from Macea (OAM) and Vatra Dornei (OAVD).

**Figure 5 molecules-30-04059-f005:**
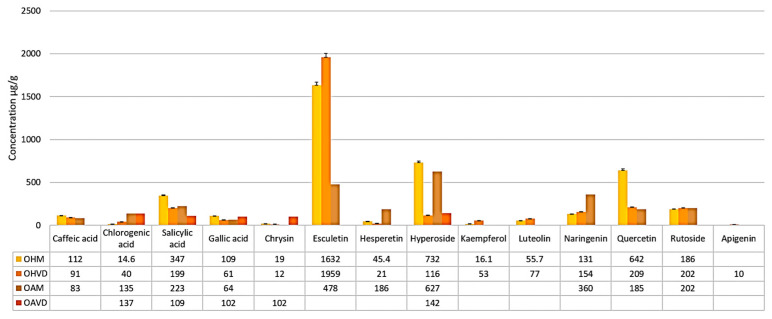
Graphical representation of phenolic acids and flavonoids identified in the hydroalcoholic extract of *O. biennis* collected from Macea (OHM) and Vatra Dornei (OHVD), and in the aqueous extract of the species collected from the same locations (OAM and OAVD, respectively).

**Table 1 molecules-30-04059-t001:** Qualitative and quantitative determination of phenolic acids and flavonoids by LC/MS in hydroalcoholic extracts of *O. biennis* L. from Macea (OHM) and Vatra Dornei (OHVD), in accordance with the standard calibration curves of standards.

Sample Compounds	OHM Sample [μg/g Dry Plant Weight]	OHVD Sample [μg/g Dry Plant Weight]
Caffeic acid	112 ± 2.8	91 ± 2.4
Chlorogenic acid	14.6 ± 0.8	40 ± 1.0
Salicylic acid	347 ± 8.7	199 ± 4.9
Gallic acid	109 ± 2.7	61 ± 1.6
Apigenin	<QL	10 ± 0.6
Chrysin	19 ± 1.1	12 ± 0.7
Esculetin	1632 ± 38.5	1959 ± 46.3
Hesperetin	45.4 ± 1.3	21± 1.2
Hyperoside	732 ± 18.3	116 ± 2.9
Kaempferol	16.1 ± 0.9	53 ± 1.4
Luteolin	55.7 ± 1.5	77 ± 2.0
Naringenin	131 ± 3.2	154 ± 3.8
Quercetin	642 ± 16.0	209 ± 5.2
Rutoside	186 ± 4.6	202 ± 5.0

Abbreviations: OHM sample—Hydroalcoholic extract, Macea sample; OHVD sample—Hydroalcoholic extract, Vatra Dornei sample; <QL = compound below quantification limit. Note: Values represent the mean ± standard deviations of three independent measurements of replicate injections of the same extract.

**Table 2 molecules-30-04059-t002:** Qualitative and quantitative determination of phenolic acids and flavonoids by LC/MS in aqueous extracts of *O. biennis* L. from Macea (OAM) and Vatra Dornei (OAVD), in accordance with the standard calibration curves of standards.

Sample Compounds	OAM Sample [μg/g Dry Plant Weight]	OAVD Sample [μg/g Dry Plant Weight]
Caffeic acid	83 ± 3.0	<QL
Chlorogenic acid	135 ± 3.3	137 ± 3.2
Salicylic acid	223 ± 5.5	109 ± 2.7
Gallic acid	102 ± 2.6	64 ± 1.7
Apigenin	<QL	<QL
Chrysin	<QL	102 ± 2.6
Esculetin	478 ± 11.9	<QL
Hesperetin	186± 4.6	<QL
Hyperoside	627 ± 15.6	142 ± 3.5
Kaempferol	<QL	<QL
Luteolin	<QL	<QL
Naringenin	360 ± 9.0	<QL
Quercetin	185 ± 4.6	<QL
Rutoside	202 ± 5.0	<QL

Abbreviations: OAM sample—Aqueous extract, Macea sample; OAVD sample—Aqueous extract, Vatra Dornei sample; <QL = compound below quantification limit. Note: Values represent the mean ± standard deviations of three independent measurements of replicate injections of the same extract.

## Data Availability

The data presented in this study are available within the article and/or Appendix A.

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
