# Peer review of "A Comparative Analysis of the Polyphenolic Content and Identification of New Compounds from Oenothera biennis L. Species from the Wild Flora"

_molecules, 2025, doi:10.3390/molecules30204059_

Round 1
Reviewer 1 Report
Comments and Suggestions for Authors
Summary & significance
The manuscript reports a comparative LC–MS analysis of polyphenols from whole-plant extracts of Oenothera biennis collected at two Romanian sites (plain vs. mountain), prepared as hydroalcoholic (70% EtOH) and aqueous infusions. The study claims detection of 5–14 polyphenols per extract and reports seven compounds “for the first time” in O. biennis (apigenin, chrysin, esculetin, hesperetin, hyperoside, luteolin, naringenin), with generally higher totals in the hydroalcoholic extract and at the lower-altitude site.
The analytical platform and basic chromatographic/mass-spectrometric settings are described, and targeted MRM transitions and calibration ranges/LOD–LOQs for the standards are provided in the Supplement.
Overall, the topic is appropriate for Molecules and potentially useful to readers interested in non-seed constituents of O. biennis. However, there are substantive issues in sampling design, validation/identification confidence, statistical treatment, and the strength of the ecological claims that require major revision before the work can be reliably interpreted.
Major comments
Sampling design, biological replication, and statistical validity
It appears that material was collected once per site during flowering, with one voucher per locality (1256, 1257). Please clarify the number of individual plants and whether extracts reflect pooled or single-individual material. Currently, only two vouchers are shown in the Supplement, suggesting one composite sample per site.
Tables report means ± SD of “three independent measurements” and site-to-site comparisons by independent t-tests. It is unclear whether these “independent measurements” are (i) replicate injections of the same extract, (ii) replicate extractions from the same bulk material, or (iii) true biological replicates (distinct plants). If they are not true biological replicates, the t-test is not appropriate. Please (a) define the replication level unambiguously, (b) provide biological replicates (n≥3 independent plants per site) or temper inferential statistics to descriptive summaries only, and (c) if hypothesis testing is retained, control the false-discovery rate across the many per-compound tests.
Novelty claim (“first time detected” in O. biennis)
The abstract and results state that seven polyphenols are reported in O. biennis for the first time. This is a central claim that must be rigorously substantiated with a systematic literature review (databases searched, date/keywords, synonyms, relevant older phytochemical surveys of O. biennis aerial parts/leaves/whole plant). At minimum, add a table cross-referencing each of the seven compounds against prior Oenothera biennis reports (or lack thereof), with full citations and plant part. As written, the claim is too global and could easily be confounded by older or non-English sources. Please also reconcile your claim explicitly with prior reports of O. biennis flavonoids in the introduction/references you already cite.
Identification confidence and method validation
The study employs targeted MRM with external standards and reports retention times, transitions, and calibration data (good). However, to support confident identification/quantification across complex matrices, please:
Provide representative overlaid chromatograms of analyte-standard vs. sample for each reported compound (hydroalcoholic and aqueous), not only one figure per extract. Ensure retention time match within acceptance windows and show at least two transitions (quantifier/qualifier) and their ion-ratio tolerances. (You list multiple transitions per analyte in Table S2—use these to strengthen IDs.)
Describe and, ideally, quantify matrix effects (post-column infusion or post-extraction spiking), recovery (spike-recovery across the working range), and intra-/inter-day precision. At present, only calibration and LOD/LOQ are given; accuracy and matrix effects are not addressed.
State clearly whether calibration was in neat solvent or matrix-matched; if the former, please justify or switch to matrix-matched curves.
For compounds reported near LOQ or flagged as “<QL” in one site but not the other (e.g., several entries in Table 2), specify decision rules for reporting presence/absence and how non-detects were handled statistically.
The Supplement spells “Apigenina”; correct all analyte names consistently across text, figures, and Supplement.
Extraction yields and back-calculation
You quantify results as μg/g dry plant weight and describe extraction ratios (0.2 g/mL for hydroalcoholic; 0.1 g/mL for aqueous). Please provide extraction yields (mg extract per g dry plant) for each sample and show the full back-calculation from measured solution concentrations to μg/g dry plant, including dilution factors. This is essential to assess comparability across solvents and sites.
Ecological/climatic interpretation
With only two sites and one collection time, it is not possible to ascribe differences in composition to altitude, temperature, precipitation, or soil pH with any statistical robustness. The manuscript currently interpolates 2020–2021 weather data and then infers ecological drivers. Please temper causal language to careful observation (“we observed differences between two sites…”) and avoid recommendations like preferred extraction source/region unless supported by a broader sampling design (multiple sites per stratum and/or multiple seasons/years). If retained, clearly label this section as exploratory.
Botanical identity & voucherization
It is good that voucher numbers and a herbarium are provided in the Supplement; however, please name the expert(s) who performed the identifications, and confirm that the deposited vouchers correspond exactly to the analyzed material (same collection date/lot). Add full locality metadata (coordinates appear in Methods, but ensure they’re on the voucher captions, too).
Presentation of quantitative results
In the main text, several interpretations rely on relative magnitudes (e.g., more compounds at the mountain site but lower concentrations overall; hydroalcoholic > aqueous). Consider adding barplots with mean±SD for each compound by site × solvent, and a stacked bar of totals, to present the pattern concisely.
Where you discuss per-compound differences, please give effect sizes (site fold-change or Δμ ± CI) rather than only p-values.
Minor comments
Language & style. Please have the manuscript reviewed by a fluent English editor. Numerous phrasing issues (e.g., long sentences; subject/verb agreement) and minor typos are present (e.g., “Apigenina” in Supplement).
Abbreviations. Define all extract codes (OHM, OHVD, OAM, OAVD) at first mention in the Abstract and again in figure/table legends for standalone clarity. Current definitions mainly appear in table headings.
Figures S4–S7. Ensure axis labels, units, and peak identities are readable at journal page width (current “Compunds analysed” typo and small type should be corrected).
Standards & transitions. For each analyte, specify which transition is the quantifier and which are qualifiers, along with allowed ion-ratio windows (±%); these details are typical in targeted LC–MS reports and will strengthen confidence.
LOD/LOQ usage. In tables, you use “<QL.” Consider adding a separate column for “Detected below LOQ” vs. “Not detected,” so readers can distinguish trace detection from absence.
Units and significant figures. The current precision (e.g., 3 significant figures for μg/g and 2 decimals for SDs) should reflect method precision; align significant figures to validation results.
Method detail. Report sample mass, extract volume, and aliquot volumes used for LC–MS injection for each extract (and any clean-up steps). The LC–MS platform is adequately described, but the sample-prep path needs the same clarity.
Plant part. Because you used whole plants, please discuss briefly how composition might differ among organs (leaves, stems, flowers, roots) and caution against extrapolating whole-plant values to specific parts used traditionally/industrially.
Specific checks against the manuscript
Claimed new detections. Clearly marked in Abstract; strengthen evidence with literature table and, ideally, product-ion spectra overlays vs. standards (where feasible) for the seven “first-time” compounds.
Aqueous-extract table. Table 2 is informative and indicates many “<QL” entries at the mountain site; decisions around detection thresholds and reporting need explicit criteria in Methods.
Targeted MRM setup and calibration. Provided in Supplement (transitions and calibration equations/LOD/LOQ). Please move a concise summary of these to Methods and leave full tables in Supplement.
Recommendation
Decision: Major Revision
Rationale: The work is potentially publishable and adds data on non-seed polyphenols of O. biennis across two environments. However, conclusions about ecological drivers are too strong for the minimal sampling; the novelty claim needs rigorous literature support; and the analytical section requires additional validation detail (matrix effects/recovery/precision) and clearer replication/statistics to support site comparisons.
Required revisions (actionable list)
Replication & statistics
Define the level of replication (injection vs. extraction vs. plant). Provide biological replicates per site or temper the inferential statistics to descriptive comparisons; if tests are retained, apply an FDR correction and report effect sizes with CIs.
Identification & validation
Add qualifier/quantifier ion details and ion-ratio criteria; include overlaid chromatograms (standard vs. sample) for each reported analyte in each matrix; state retention-time tolerances; and report matrix-matched calibration, recovery, and intra/inter-day precision.
Novelty substantiation
Provide a Supplementary table summarizing prior O. biennis reports by compound/plant part, database search strategy, and dates; adjust “first-time” language as needed.
Extraction yields & calculations
Report percent yields for each extract and show the equations used to convert instrumental concentrations to μg/g dry plant, including all dilutions.
Ecological claims
Recast causal language regarding altitude/temperature/precipitation as observational; move “recommendation” about preferred source region/solvent to a cautious perspective paragraph unless supported by expanded sampling.
Botanical/metadata clarity
Name the identifier(s), confirm vouchers correspond to analyzed lots, and add full locality/collection metadata in the caption or a data table.
Editing & presentation
Professional English edit; correct “Apigenina” and other typos; ensure all figure/table labels are legible and units consistent; add a compact figure summarizing per-compound abundances by site × solvent.
Author Response
Major comments
Sampling design, biological replication, and statistical validity
It appears that material was collected once per site during flowering, with one voucher per locality (1256, 1257). Please clarify the number of individual plants and whether extracts reflect pooled or single-individual material. Currently, only two vouchers are shown in the Supplement, suggesting one composite sample per site.
Reply: This is a valuable remark, therefore we added the following clarification to the Materials and Methods section:
„Several individuals belonging to the same population were collected from each location, with one specimen per population preserved as a voucher, and the rest used for the preparation of extracts. The number of individuals collected per location varied depending on availability and plant biomass. Collection was carried out in compliance with the legislation, research regulations in traditional medicine [106-108], and scientific literature recommendations regarding weather conditions at the time of sampling—warm weather, no precipitation, and ontogenetic stage—at maturity, respectively during the flowering period [109].”
Tables report means ± SD of “three independent measurements” and site-to-site comparisons by independent t-tests. It is unclear whether these “independent measurements” are (i) replicate injections of the same extract, (ii) replicate extractions from the same bulk material, or (iii) true biological replicates (distinct plants). If they are not true biological replicates, the t-test is not appropriate. Please (a) define the replication level unambiguously, (b) provide biological replicates (n≥3 independent plants per site) or temper inferential statistics to descriptive summaries only, and (c) if hypothesis testing is retained, control the false-discovery rate across the many per-compound tests.
Reply: The three independent measurements are replicate injections of the same extract. We introduced a paragraph in section 4.7 Statistical analysis from Materials and Methods to clarify this aspect, and eliminated the T-test. Inferential statistics were tempered to descriptive summaries.
Novelty claim (“first time detected” in O. biennis)
The abstract and results state that seven polyphenols are reported in O. biennis for the first time. This is a central claim that must be rigorously substantiated with a systematic literature review (databases searched, date/keywords, synonyms, relevant older phytochemical surveys of O. biennis aerial parts/leaves/whole plant). At minimum, add a table cross-referencing each of the seven compounds against prior Oenothera biennis reports (or lack thereof), with full citations and plant part. As written, the claim is too global and could easily be confounded by older or non-English sources. Please also reconcile your claim explicitly with prior reports of O. biennis flavonoids in the introduction/references you already cite.
Reply: We’ve inserted Table S1 in the Supplementary material with a comprehensive overview of previous studies on O. biennis phytochemical composition, in chronological order per plant part, including vegetative organs (leaves, stem, roots), reproductive parts (seeds, seed derivates), whole plant and sprouts. In Discussion section we align our results with prior reports of O. biennis for each compound. Section 4.6 was added to Materials and Methods detailing the literature review methodology.
Initially we did not include a systematic literature review for brevity and focus, as this is meant to be an original research paper and not a review paper. However, we are fully aware that our claim must be done with careful consideration of the scientific literature. It is a species that we have studied for several years and regularly updated our knowledge of new publications on this subject. Therefore, the references from Table S1 were gathered from several databases over the course of over 5 years. Prior to our submission of the manuscript we have checked the online databases for possible new publications regarding each compound for which we claim first report. It is to the best of our knowledge that there are no previous mentions in the scientific literature to date of these particular seven flavonoids for O. biennis species.
Identification confidence and method validation
The study employs targeted MRM with external standards and reports retention times, transitions, and calibration data (good). However, to support confident identification/quantification across complex matrices, please:
Provide representative overlaid chromatograms of analyte-standard vs. sample for each reported compound (hydroalcoholic and aqueous), not only one figure per extract. Ensure retention time match within acceptance windows and show at least two transitions (quantifier/qualifier) and their ion-ratio tolerances. (You list multiple transitions per analyte in Table S2—use these to strengthen IDs.)
Reply: Thank you for this observation. Indeed, overlapping chromatograms would provide a better visual indication of retention time correspondence, but LC/MS software does not allow the same type of overlap as in a standard HPLC. For this reason, a table showing the retention times of the standards in parallel with those of the corresponding components separated from the studied extracts has been included in the supplementary material. The data confirms the correspondence of these parameters.
We have also included in the supplementary material the MS spectra of the separated components, confirming the primary and secondary transitions mentioned in Table S3.
Describe and, ideally, quantify matrix effects (post-column infusion or post-extraction spiking), recovery (spike-recovery across the working range), and intra-/inter-day precision. At present, only calibration and LOD/LOQ are given; accuracy and matrix effects are not addressed.
Reply: Indeed, the matrix effect can be significant in biological samples. It is significant when biological fluids are used. In the case of alcoholic or even aqueous extracts, the extraction of high molecular weight compounds that could give a significant matrix effect is unlikely due to the solvating properties and small size of the solvent molecules. As a result, the matrix effect can be considered insignificant in this case. The coelution of small compounds in LC/MS is usually resolved by the selectivity of specific transitions.
State clearly whether calibration was in neat solvent or matrix-matched; if the former, please justify or switch to matrix-matched curves.
Reply: Given that the matrix is the extraction solvent used, we can say that the standards were solubilized in the same type of solvent, so the calibration curves were made in pure solvent, which can also be considered a matrix.
For compounds reported near LOQ or flagged as “<QL” in one site but not the other (e.g., several entries in Table 2), specify decision rules for reporting presence/absence and how non-detects were handled statistically.
The Supplement spells “Apigenina”; correct all analyte names consistently across text, figures, and Supplement.
Reply: Adjusted and double-checked that all compound names are correct and consistently spelled across manuscript and Supplement materials. Thank you for observing this.
Extraction yields and back-calculation
You quantify results as μg/g dry plant weight and describe extraction ratios (0.2 g/mL for hydroalcoholic; 0.1 g/mL for aqueous). Please provide extraction yields (mg extract per g dry plant) for each sample and show the full back-calculation from measured solution concentrations to μg/g dry plant, including dilution factors. This is essential to assess comparability across solvents and sites.
Reply: The extractions yields are 20 mg/g dry plant for alcoholic extracts and 10 mg/g dry plant for aqueous extracts. We have back-calculated from extract to dry plant the concentrations counting the extraction ratio. We have added a paragraph in the Materials and Methods to clarify this aspect.
Ecological/climatic interpretation
With only two sites and one collection time, it is not possible to ascribe differences in composition to altitude, temperature, precipitation, or soil pH with any statistical robustness. The manuscript currently interpolates 2020–2021 weather data and then infers ecological drivers. Please temper causal language to careful observation (“we observed differences between two sites…”) and avoid recommendations like preferred extraction source/region unless supported by a broader sampling design (multiple sites per stratum and/or multiple seasons/years). If retained, clearly label this section as exploratory.
Reply: Thank you for your remark. We have reformulated the respective paragraphs from the Abstract, Discussion and Conclusion sections as follows:
“Climatic parameters mapped using QGIS, and results were analyzed in relation to altitude and pedoclimatic factors. Higher qualitative and quantitative values of poly-phenols were observed in the hydroalcoholic extract obtained from individuals collected from the plain area in relation to the analyzed environmental factors.”
“Moreover, we observed differences in the polyphenolic content of hydroalcoholic and aqueous extracts, as well as between the two sites with characteristic plain and mountain pedoclimatic conditions. The quantitative and qualitative values were higher for hydroalcoholic extracts from O. biennis plants originating from low-altitude areas with a warm climate, high average annual temperatures, low cumulative precipitation, and basic soil, such as in the Macea area.”
“The analysis of the polyphenolic content of O. biennis species collected from two different regions of the wild flora of Romania has revealed quantitative and qualitative differences between the two sites characterized by distinct pedoclimatic factors (plain and mountain), and between the two types of extract (hydroalcoholic and aqueous).”
Botanical identity & voucherization
It is good that voucher numbers and a herbarium are provided in the Supplement; however, please name the expert(s) who performed the identifications, and confirm that the deposited vouchers correspond exactly to the analyzed material (same collection date/lot). Add full locality metadata (coordinates appear in Methods, but ensure they’re on the voucher captions, too).
Reply: The required information was provided on the labels attached to the herbarium files, however, we understand that the quality of the images and the fact that the labels are written in Romanian may hinder the reader's understanding so we translated the labels in English and increased the resolution of the Figures S1 and S2.
Presentation of quantitative results
In the main text, several interpretations rely on relative magnitudes (e.g., more compounds at the mountain site but lower concentrations overall; hydroalcoholic > aqueous). Consider adding barplots with mean±SD for each compound by site × solvent, and a stacked bar of totals, to present the pattern concisely.
Reply: Visual representations with barplots ±SD were added as Figures 3, 4, and 5, showing compound variations between locations for each type of extract, and between site and extract types together. Additionally, we specify the concentrations ±SD for all compound mentions in text. We hope this provides a more clear view of the results.
Where you discuss per-compound differences, please give effect sizes (site fold-change or Δμ ± CI) rather than only p-values.
Reply: Thank you for your observation. However, during the revision we have eliminated the t-test and tempered our language to descriptive summaries. Also, the p-values were previously mentioned only in tables, so we are not sure if any additional change is needed.
Minor comments
Language & style. Please have the manuscript reviewed by a fluent English editor. Numerous phrasing issues (e.g., long sentences; subject/verb agreement) and minor typos are present (e.g., “Apigenina” in Supplement).
Reply: Manuscript language was revised and “Apigenin” was corrected in Supplement material.
Abbreviations. Define all extract codes (OHM, OHVD, OAM, OAVD) at first mention in the Abstract and again in figure/table legends for standalone clarity. Current definitions mainly appear in table headings.
Reply: We introduced a first mention and definition of each code in the Abstract, and in the table headings, as the abbreviations were previously explained only in table notes, and at their first mention in Results section.
Figures S4–S7. Ensure axis labels, units, and peak identities are readable at journal page width (current “Compunds analysed” typo and small type should be corrected).
Reply: We have increased the size and resolution of the individual images previously presented in Figures S4-S7, now S4-S12. “Compunds analysed” was deemed redundant and deleted.
Standards & transitions. For each analyte, specify which transition is the quantifier and which are qualifiers, along with allowed ion-ratio windows (±%); these details are typical in targeted LC–MS reports and will strengthen confidence.
Reply: We have added in the Materials and Methods the following clarification: „The main transition was used for quantification, and both the main and secondary transitions were used for identification.”
LOD/LOQ usage. In tables, you use “<QL.” Consider adding a separate column for “Detected below LOQ” vs. “Not detected,” so readers can distinguish trace detection from absence.
Units and significant figures. The current precision (e.g., 3 significant figures for μg/g and 2 decimals for SDs) should reflect method precision; align significant figures to validation results.
Reply: The results being back-calculated we used 1 significant figure for data and 2 significant figures for SDs.
Method detail. Report sample mass, extract volume, and aliquot volumes used for LC–MS injection for each extract (and any clean-up steps). The LC–MS platform is adequately described, but the sample-prep path needs the same clarity.
Reply: We have added in the Materials and Methods the following clarification: „Each sample/extract was diluted 1 ml to 10 ml with methanol, and aliquots of 10 μL were injected into LC/MS. The diluted samples were filtered through 0,45 μm PFTE membrane filters.”
Plant part. Because you used whole plants, please discuss briefly how composition might differ among organs (leaves, stems, flowers, roots) and caution against extrapolating whole-plant values to specific parts used traditionally/industrially.
Reply: Excellent remark! Thank you for your suggestion. We have introduced in Discussion a paragraph that addresses this aspect and aligns our results with previous publications.
Specific checks against the manuscript
Claimed new detections. Clearly marked in Abstract; strengthen evidence with literature table and, ideally, product-ion spectra overlays vs. standards (where feasible) for the seven “first-time” compounds.
Reply: A literature review table was added in Supplement, and results were presented in alignment with the references cited.
Aqueous-extract table. Table 2 is informative and indicates many “<QL” entries at the mountain site; decisions around detection thresholds and reporting need explicit criteria in Methods.
Targeted MRM setup and calibration. Provided in Supplement (transitions and calibration equations/LOD/LOQ). Please move a concise summary of these to Methods and leave full tables in Supplement.
Recommendation
Decision: Major Revision
Rationale: The work is potentially publishable and adds data on non-seed polyphenols of O. biennis across two environments. However, conclusions about ecological drivers are too strong for the minimal sampling; the novelty claim needs rigorous literature support; and the analytical section requires additional validation detail (matrix effects/recovery/precision) and clearer replication/statistics to support site comparisons.
Required revisions (actionable list)
Replication & statistics
Define the level of replication (injection vs. extraction vs. plant). Provide biological replicates per site or temper the inferential statistics to descriptive comparisons; if tests are retained, apply an FDR correction and report effect sizes with CIs.
Identification & validation
Add qualifier/quantifier ion details and ion-ratio criteria; include overlaid chromatograms (standard vs. sample) for each reported analyte in each matrix; state retention-time tolerances; and report matrix-matched calibration, recovery, and intra/inter-day precision.
Novelty substantiation
Provide a Supplementary table summarizing prior O. biennis reports by compound/plant part, database search strategy, and dates; adjust “first-time” language as needed.
Extraction yields & calculations
Report percent yields for each extract and show the equations used to convert instrumental concentrations to μg/g dry plant, including all dilutions.
Ecological claims
Recast causal language regarding altitude/temperature/precipitation as observational; move “recommendation” about preferred source region/solvent to a cautious perspective paragraph unless supported by expanded sampling.
Botanical/metadata clarity
Name the identifier(s), confirm vouchers correspond to analyzed lots, and add full locality/collection metadata in the caption or a data table.
Editing & presentation
Professional English edit; correct “Apigenina” and other typos; ensure all figure/table labels are legible and units consistent; add a compact figure summarizing per-compound abundances by site × solvent.
Reply: Thank you for your kind constructive criticism. We hope we have addressed all suggestions accordingly.
Reviewer 2 Report
Comments and Suggestions for Authors
Oenothera biennis L. is a species of high pharmacological interest. This manuscript studies the polyphenolic profile of the O. biennis species from two Romanian regions with different altitudes and pedoclitmatic conditions (plains and mountains) to determine the impact of these factors on polyphenol content. Highlights include seven new polyphenols for this species and significant differences between the studied populations due to the different environmental conditions.
This is a well-conceived work that presents relevant information about an interesting plant genetic resource from Romania. The manuscript is well-written and reflects the experience of the researchers involved in the project.
The introduction provides a state-of-the-art overview of O. biennis regarding its composition. This section is well-conceived because it provides insight into what is known about this species and what remains to be learned.
The objective is clear, and the methodology used is well-presented and meets the necessary requirements. I emphasize that the authors collected sample material from the studied populations to include it in the herbarium. This process is sometimes not performed by researchers working in phytochemistry, making the work unreplicable.
Regarding the format of the work, the figures presented are of high quality and reflect interesting information about the climate of the studied populations. The tables of results are also well presented. The results are correctly presented, and the discussion uses up-to-date bibliography.
The Conclusion section of the work that clearly needs to be improved. The authors present a summary of the results in this section, but the authors should say more here.
Author Response
Oenothera biennis L. is a species of high pharmacological interest. This manuscript studies the polyphenolic profile of the O. biennis species from two Romanian regions with different altitudes and pedoclitmatic conditions (plains and mountains) to determine the impact of these factors on polyphenol content. Highlights include seven new polyphenols for this species and significant differences between the studied populations due to the different environmental conditions.
This is a well-conceived work that presents relevant information about an interesting plant genetic resource from Romania. The manuscript is well-written and reflects the experience of the researchers involved in the project.
The introduction provides a state-of-the-art overview of O. biennis regarding its composition. This section is well-conceived because it provides insight into what is known about this species and what remains to be learned.
The objective is clear, and the methodology used is well-presented and meets the necessary requirements. I emphasize that the authors collected sample material from the studied populations to include it in the herbarium. This process is sometimes not performed by researchers working in phytochemistry, making the work unreplicable.
Regarding the format of the work, the figures presented are of high quality and reflect interesting information about the climate of the studied populations. The tables of results are also well presented.
The results are correctly presented, and the discussion uses up-to-date bibliography.
The Conclusion section of the work that clearly needs to be improved. The authors present a summary of the results in this section, but the authors should say more here.
Reply: Thank you for supporting our work and the publication of our manuscript. We hope the revised Conclusion is an improvement.
Reviewer 3 Report
Comments and Suggestions for Authors
The manuscript, molecules-3877113, is well-written. It is concise yet comprehensive, providing the essential information. The text has been meticulously edited and proofread to ensure accuracy. I found no factual errors and appreciate the discussion. However, I have other significant concerns regarding this submission. In the title, the authors used the term "polyphenolic profile". This expression indicates the presence of a set of phenolic compounds along with their derivatives (methyl, glycoside, etc.). My plant phenolic profiles contain 60–100 compounds. The authors presented a maximum of 14 compounds. A discrepancy exists between the title and the content of the manuscript. The authors (of whom there are nine) should have considered identifying additional compounds in these plants, including “climate-sensitive” compounds, which would have given the work a broader context.
I am dissatisfied after reading the manuscript. Although it is quite extensive (15 pages), I believe it would be more appropriate as a short communication, as it currently contains insufficient data set. I would like to request that the editor and authors consider reclassifying the manuscript.
Additionally, I would like to suggest one minor correction that would improve the manuscript. The climate maps contain city names. Currently, they are blurred. In addition to improving the resolution, I would use contrasting font colors (dark names on a light background and vice versa).
September 21, 2025
Author Response
The manuscript, molecules-3877113, is well-written. It is concise yet comprehensive, providing the essential information. The text has been meticulously edited and proofread to ensure accuracy. I found no factual errors and appreciate the discussion. However, I have other significant concerns regarding this submission. In the title, the authors used the term "polyphenolic profile". This expression indicates the presence of a set of phenolic compounds along with their derivatives (methyl, glycoside, etc.). My plant phenolic profiles contain 60–100 compounds. The authors presented a maximum of 14 compounds. A discrepancy exists between the title and the content of the manuscript. The authors (of whom there are nine) should have considered identifying additional compounds in these plants, including “climate-sensitive” compounds, which would have given the work a broader context.
Reply: We have tempered the title and replaced “polyphenolic profile” with “polyphenolic content”, which we hope provides a better alignment with the results presented in the manuscript. The terminology was replaced throughout the whole manuscript.
I am dissatisfied after reading the manuscript. Although it is quite extensive (15 pages), I believe it would be more appropriate as a short communication, as it currently contains insufficient data set. I would like to request that the editor and authors consider reclassifying the manuscript.
Reply: We are sorry to hear about your dissatisfaction and have carefully considered this comment. In the light of all the changes brought to the manuscript during the revision, the comparative analysis between 2 types of extract obtained from 2 populations originating from sites characterized by vastly different pedoclimatic conditions that were analyzed as well, and the report of new flavonoids for this species, backed by a comprehensive bibliographic study, we hope it can be accepted as an original research article, and wait to hear the editor’s opinion as well.
Additionally, I would like to suggest one minor correction that would improve the manuscript. The climate maps contain city names. Currently, they are blurred. In addition to improving the resolution, I would use contrasting font colors (dark names on a light background and vice versa).
Reply: In order to keep as much as possible from the original resolution while shrinking the images to fit the Journal’s page, we inserted them individually. The names were contrasted using Adobe Photoshop as much as possible without impacting the interpolation in the background. A complete remake of the maps would have exceeded the revision deadline, so we hope that the current adjustments are sufficient to clarify the mentioned issues.
Reviewer 4 Report
Comments and Suggestions for Authors
The article by Bona and colleagues analyzes the polyphenolic profile of Oenothera biennis L. collected in Romania, both from lowland and mountain areas, highlighting the influence of pedoclimatic conditions on polyphenol composition. Oenothera biennis is a biennial species native to North America, traditionally used in medicine and as a dietary supplement. For both areas, hydroalcoholic and aqueous extracts of the plant were analyzed, revealing differences in polyphenolic content depending on the location and the type of extract. The article also describes the methodology used for the statistical analysis of the collected data.
The manuscript is well written and pleasant to read. The introduction is well structured and provides a comprehensive overview—perhaps a bit too long, but very detailed. The text is well organized and offers a thorough analysis of the study’s results. The document includes the appropriate bibliographic references.
Author Response
The article by Bona and colleagues analyzes the polyphenolic profile of Oenothera biennis L. collected in Romania, both from lowland and mountain areas, highlighting the influence of pedoclimatic conditions on polyphenol composition. Oenothera biennis is a biennial species native to North America, traditionally used in medicine and as a dietary supplement. For both areas, hydroalcoholic and aqueous extracts of the plant were analyzed, revealing differences in polyphenolic content depending on the location and the type of extract. The article also describes the methodology used for the statistical analysis of the collected data.
The manuscript is well written and pleasant to read. The introduction is well structured and provides a comprehensive overview—perhaps a bit too long, but very detailed. The text is well organized and offers a thorough analysis of the study’s results. The document includes the appropriate bibliographic references.
Reply: Thank you for supporting our work and the publication of our manuscript.
Reviewer 5 Report
Comments and Suggestions for Authors
The manuscript submitted for review examines the polyphenolic profile of the O. biennis species collected from the wild flora of Romania, from two regions with different altitudes and pedoclimatic conditions (plains and mountains), in order to determine the impact of these factors on the polyphenol content, both qualitatively and quantitatively.
The paper is relevant to the topic, has all the necessary sections and the study has a well structured methodology.
Remark:
- The polyphenols could be presented in groups (Table 1, Line 175-176; Table 2, Line 193-194).
- The conclusion need to be supplemented with the possibilities for future work.
Accepted after minor revision.
Author Response
The manuscript submitted for review examines the polyphenolic profile of the O. biennis species collected from the wild flora of Romania, from two regions with different altitudes and pedoclimatic conditions (plains and mountains), in order to determine the impact of these factors on the polyphenol content, both qualitatively and quantitatively.
The paper is relevant to the topic, has all the necessary sections and the study has a well structured methodology.
Remark:
The polyphenols could be presented in groups (Table 1, Line 175-176; Table 2, Line 193-194).
The conclusion need to be supplemented with the possibilities for future work.
Accepted after minor revision.
Reply: The polyphenols are now presented in the heading of Table 1 and 2 as their respective class - phenolic acids and flavonoids.
The following suggestions were added to the Conclusion:
“Future studies involving multiple sites, bigger sampling over a longer period, and soil analysis will be necessary to better understand the extent these ecological factors have over the polyphenolic content of this species.”
Thank you for supporting our work and the publication of our manuscript.
Round 2
Reviewer 1 Report
Comments and Suggestions for Authors
Summary & significance
The paper compares LC–MS profiles of polyphenols in whole-plant extracts of Oenothera biennis from two Romanian sites (plain vs mountain), using hydroalcoholic and aqueous preparations. The revision adds a literature overview and clearer methods, and reframes ecological claims more cautiously. The topic fits Molecules and the dataset can be useful for readers interested in non-seed constituents of O. biennis. Still, several analytical-validation and reporting gaps remain before the results can be relied upon.
What the authors improved
Sampling description. Methods now state that several individuals were collected per site (one voucher kept, others used for extracts), with timing, conditions and compliance references. This addresses the ambiguity about single-plant vs pooled material.
Abbreviations & framing. Extract codes (OHM, OHVD, OAM, OAVD) are defined in the Abstract; the text consistently introduces the two sites and extract types.
Extraction yields & back-calculation. The Methods now report extraction ratios and yields (20 mg/g for 70% EtOH; 10 mg/g for aqueous) and note back-calculation to μg/g dry plant.
Statistics toned down. Tables retain means ± SD of three measurements; inferential t-tests have been removed and the narrative is descriptive (appropriate given replication level). (Tables show the “three independent measurements” note).
Visualization. New barplots summarise per-compound results by site × extract (Figures 3–5).
Systematic literature check. The Supplement now includes a long, structured Table S1 of prior phytochemical reports across plant parts and preparations; the main text discusses alignment and novelty claims referencing this table.
Analytical details expanded. The Supplement lists gradient program (Table S2), transitions and retention times (Table S3), and calibration/LOD-LOQ information (Table S4); the main Methods now state that chromatograms (S4–S12) and MS spectra (S13–S19) are provided.
Ecological interpretation tempered. Abstract/Discussion have been rephrased to “observed differences” linked to pedoclimatic contrasts rather than causal claims.
What still needs work (major points)
Identification confidence & validation are still insufficient in the manuscript
What improved: The authors added transitions and retention times (S3), calibration/LOD/LOQ (S4), and say spectra and chromatograms are shown (S13–S19, S4–S12).
Still needed:
Ion-ratio criteria & tolerances. The text states “main transition used for quant, main + secondary for ID,” but it does not give qualifier/quantifier ion-ratio windows (e.g., ±20–30%) or RT-match tolerances. Please add per-analyte ion-ratio criteria and acceptance limits.
Matrix effects & recovery. The manuscript still lacks objective assessments (post-extraction spikes, matrix-matched calibration, recovery % across range, intra/inter-day precision). The reply argues matrix effects are “insignificant” in these extracts and treats pure solvent as “the matrix,” which is not acceptable as validation in complex plant matrices. Please quantify matrix effects (at least a simple post-extraction spike vs neat), report recoveries, and confirm precision.
Presence/absence near LOQ. Define decision rules for “<QL” vs “ND,” how values below LOQ were treated in summaries, and the RT/ion-ratio criteria required to count a compound as “present” without quantitation. (Tables show <QL but no rule is stated in Methods.)
Novelty claim still needs a tighter audit trail
The Abstract and Discussion continue to claim seven flavonoids are “newly reported for this species.” The new Table S1 is helpful, and the Introduction mentions a review through August 2025, but the search protocol (databases, date of final search, keywords/synonyms, inclusion of non-English sources) should be summarised in Methods and a one-row-per-compound novelty table should explicitly show “previous reports in O. biennis (Y/N) by plant part, ref(s).” This will future-proof the claim.
Replication language
The paper now (correctly) avoids inferential tests. Please state plainly in Methods/Statistics that the reported SD reflects technical (injection) replicates of a single pooled extract (not biological replicates) so readers don’t over-interpret error bars. (Tables currently say “three independent measurements” but “independent” is ambiguous to non-specialists.)
Figure/caption consistency and a few presentation issues
Figure 4 caption calls it an “aqueous extract” figure but refers to OHM/OHVD (the hydroalcoholic codes) instead of OAM/OAVD. Please correct.
Significant figures policy vs tables. The response says “1 significant figure for data and 2 for SDs,” but Table 1 still reports, e.g., 14.6 ± 0.81 and 55.7 ± 1.54, which don’t match that rule. Please align all tables to a consistent, method-precision-based sig-fig policy.
Ensure all axis labels/units remain legible in the final PDF, and that the new figures (3–5) use the same abbreviations defined in the Abstract.
Botanical/voucher clarity in the paper body
The Supplement shows vouchers and the Methods list coordinates and altitudes for both sites. In the main text, add a brief sentence identifying who confirmed the species ID and explicitly state that the voucher corresponds to the analyzed lots (same collection/date). This keeps crucial provenance in the Article itself, not only in image captions.
Minor/technical notes
Chromatogram overlays. If the LC/MS software cannot overlay as HPLC, consider exporting traces and composing overlays externally; at minimum, include paired RT tables (standard vs sample) for every analyte and matrix (you say S4 shows this)—ensure these are complete and clearly labeled.
Method traceability. The LC/MS section is clearer now (column, gradient, ESI mode, flow, temps). Consider adding RT tolerance used for ID (e.g., ±0.2 min) and noting which transition is the quantifier for each analyte (S3 lists transitions).
Ecological language. The toned-down phrasing is appropriate; keep it strictly observational in Abstract and Conclusions.
Required revisions (actionable checklist)
Matrix effects & recovery/precision
Quantify matrix effects (e.g., post-extraction spike vs neat) for representative analytes in both matrices; report %ME.
Provide spike-recovery (%) across the working range, plus intra-/inter-day precision for a pooled extract.
If matrix effects are non-negligible, use matrix-matched calibration or internal standards and revise concentrations accordingly.
Identification criteria
Specify, per analyte, quantifier and qualifier transitions, ion-ratio tolerance, and RT tolerance. Confirm that all reported compounds meet these acceptance criteria in each matrix.
Presence/absence policy near LOQ
Add explicit decision rules for “<QL,” “ND,” and “Detected <LOQ” in Methods; state how such values are handled in summaries/plots. Apply consistently across Tables 1–2 and Figures 3–5.
Novelty substantiation
In Methods, summarise the systematic search protocol (databases, final search date, keywords/synonyms, languages).
Add a compact Supplementary novelty table (one row per of the 7 claimed compounds) showing: prior reports in O. biennis (Y/N), plant part, citation(s). Adjust claim language if any conflicts emerge.
Replication clarification
State explicitly that error bars/SD reflect technical replicate injections of a pooled extract (not biological replicates). Consider adding this note to figure captions.
Presentation fixes
Correct Figure 4 caption to reference OAM/OAVD (aqueous) rather than OHM/OHVD.
Harmonize significant figures across all tables to your declared policy (or revise the policy to reflect actual method precision).
In the Article body, add a one-sentence species-ID provenance (identifier name/affiliation) and state that vouchers correspond to analyzed lots.
Author Response
What still needs work (major points)
Identification confidence & validation are still insufficient in the manuscript
What improved: The authors added transitions and retention times (S3), calibration/LOD/LOQ (S4), and say spectra and chromatograms are shown (S13–S19, S4–S12).
Still needed:
Ion-ratio criteria & tolerances. The text states “main transition used for quant, main + secondary for ID,” but it does not give qualifier/quantifier ion-ratio windows (e.g., ±20–30%) or RT-match tolerances. Please add per-analyte ion-ratio criteria and acceptance limits.
Reply: We have included the limits of ± 1.5 min for retention time and ± 1% for transitions in the corresponding tables (S6–S9) from the Supplementary Materials.
Matrix effects & recovery. The manuscript still lacks objective assessments (post-extraction spikes, matrix-matched calibration, recovery % across range, intra/inter-day precision). The reply argues matrix effects are “insignificant” in these extracts and treats pure solvent as “the matrix,” which is not acceptable as validation in complex plant matrices. Please quantify matrix effects (at least a simple post-extraction spike vs neat), report recoveries, and confirm precision.
Reply: Materials and Methods were completed with the following statement:
“To determine the matrix effect, standards were added to each extract so that the added concentration was twice that determined in the extract. LC/MS analyses were performed identically to the initial analyses. The same calibration curves were used for the calculations as for the initial determinations.
Recovery was determined as the percentage of the ratio of the concentration determined in the sample with standard addition and the standard addition concentration. The recoveries obtained were between 148 and 151 %, which corresponds to twice the concentration added to the samples.
The matrix effect was determined as the percentage of the ratio between the concentration of the spiked sample and the sum of the concentrations of the unspiked sample and the concentration of the added standard. The percentages obtained range between 99.5 and 100.5%. Therefore, the matrix effect is very low and can be considered insignificant.
The determinations for the matrix effect were performed in triplicate, and the recovery and matrix effect percentages mentioned are calculated as the average of the three determinations. The relative standard deviation of the three determinations for each sample and each separate and quantified component is below 10%.”
Presence/absence near LOQ. Define decision rules for “<QL” vs “ND,” how values below LOQ were treated in summaries, and the RT/ion-ratio criteria required to count a compound as “present” without quantitation. (Tables show <QL but no rule is stated in Methods.)
Reply: Definitions for “ND” and “<QL” were added to Materials and Methods.
“ND” is defined as “none detected” or “under detection limit”. “<QL” is defined as a component that can be identified (at a concentration greater than the detection limit), but which cannot be quantified within a reasonable error limit.
Novelty claim still needs a tighter audit trail
The Abstract and Discussion continue to claim seven flavonoids are “newly reported for this species.” The new Table S1 is helpful, and the Introduction mentions a review through August 2025, but the search protocol (databases, date of final search, keywords/synonyms, inclusion of non-English sources) should be summarised in Methods and a one-row-per-compound novelty table should explicitly show “previous reports in O. biennis (Y/N) by plant part, ref(s).” This will future-proof the claim.
Reply: Bibliographic study protocol was included in section 4.6 of Materials and Methods, mentioning the points required above. We consulted major scientific databases on various kinds of publications, both at international and national levels, in different languages (such as English, Romanian, German, and Japanese with abstract translation). To ensure a thorough and systematic approach, we utilized various combinations of keywords, including the species’ name + each compound we claim first report. We previously wrote that «The bibliographic study encompassed all database search results published before August 29, 2025 » because that was the date of the final search, right before the submission of the manuscript. However, given the new search performed during the revisions, we updated our mention to October 6, 2025.
Despite multiple bibliographic studies, we have found no previous reports in O. biennis species, regardless of plant part, of the particular 7 flavonoids we mention in our paper as “first report”. Therefore, we added Table S2, explicitly showing previous reports in O. biennis by plant part and type of extract of all compounds identified in this study.
Replication language
The paper now (correctly) avoids inferential tests. Please state plainly in Methods/Statistics that the reported SD reflects technical (injection) replicates of a single pooled extract (not biological replicates) so readers don’t over-interpret error bars. (Tables currently say “three independent measurements” but “independent” is ambiguous to non-specialists.)
Reply: We completed Tables 1 and 2 notes, specifying that SDs are “three independent measures of replicate injections of the same extract”.
Figure/caption consistency and a few presentation issues
Figure 4 caption calls it an “aqueous extract” figure but refers to OHM/OHVD (the hydroalcoholic codes) instead of OAM/OAVD. Please correct.
Significant figures policy vs tables. The response says “1 significant figure for data and 2 for SDs,” but Table 1 still reports, e.g., 14.6 ± 0.81 and 55.7 ± 1.54, which don’t match that rule. Please align all tables to a consistent, method-precision-based sig-fig policy.
Ensure all axis labels/units remain legible in the final PDF, and that the new figures (3–5) use the same abbreviations defined in the Abstract.
Reply: Thank you for your observation. We corrected Figure 4. Additionally, we verified Figures 3-5 to ensure that they use the same abbreviations defined in the Abstract both in the picture itself and in their captions. Figure size was increased by 1%, and high-resolution versions were included in the uploaded .zip folder.
Significant figures were adjusted across all tables.
Botanical/voucher clarity in the paper body
The Supplement shows vouchers and the Methods list coordinates and altitudes for both sites. In the main text, add a brief sentence identifying who confirmed the species ID and explicitly state that the voucher corresponds to the analyzed lots (same collection/date). This keeps crucial provenance in the Article itself, not only in image captions.
Reply: We explicitly state in the Materials and Methods section 4.1. the required information.
Minor/technical notes
Chromatogram overlays. If the LC/MS software cannot overlay as HPLC, consider exporting traces and composing overlays externally; at minimum, include paired RT tables (standard vs sample) for every analyte and matrix (you say S4 shows this)—ensure these are complete and clearly labeled.
Method traceability. The LC/MS section is clearer now (column, gradient, ESI mode, flow, temps). Consider adding RT tolerance used for ID (e.g., ±0.2 min) and noting which transition is the quantifier for each analyte (S3 lists transitions).
Ecological language. The toned-down phrasing is appropriate; keep it strictly observational in Abstract and Conclusions.
Reply: We hope the added information in the Tables and Supplementary addresses the issue in a satisfactory manner. The ecological language in the Abstract and Conclusion was further adjusted to simple observations.
Required revisions (actionable checklist)
Matrix effects & recovery/precision
Quantify matrix effects (e.g., post-extraction spike vs neat) for representative analytes in both matrices; report %ME.
Provide spike-recovery (%) across the working range, plus intra-/inter-day precision for a pooled extract.
If matrix effects are non-negligible, use matrix-matched calibration or internal standards and revise concentrations accordingly.
Reply: Added.
Identification criteria
Specify, per analyte, quantifier and qualifier transitions, ion-ratio tolerance, and RT tolerance. Confirm that all reported compounds meet these acceptance criteria in each matrix.
Reply: Added.
Presence/absence policy near LOQ
Add explicit decision rules for “<QL,” “ND,” and “Detected <LOQ” in Methods; state how such values are handled in summaries/plots. Apply consistently across Tables 1–2 and Figures 3–5.
Reply: Added.
Novelty substantiation
In Methods, summarise the systematic search protocol (databases, final search date, keywords/synonyms, languages).
Reply: Summary added.
Add a compact Supplementary novelty table (one row per of the 7 claimed compounds) showing: prior reports in O. biennis (Y/N), plant part, citation(s). Adjust claim language if any conflicts emerge.
Reply: Table S2 added.
Replication clarification
State explicitly that error bars/SD reflect technical replicate injections of a pooled extract (not biological replicates). Consider adding this note to figure captions.
Reply: Adjusted.
Presentation fixes
Correct Figure 4 caption to reference OAM/OAVD (aqueous) rather than OHM/OHVD.
Harmonize significant figures across all tables to your declared policy (or revise the policy to reflect actual method precision).
In the Article body, add a one-sentence species-ID provenance (identifier name/affiliation) and state that vouchers correspond to analyzed lots.
Reply: Adjusted and completed.